# Current Update on Biomarkers for Detection of Cancer: Comprehensive Analysis

**DOI:** 10.3390/vaccines10122138

**Published:** 2022-12-13

**Authors:** Ankur Kaushal, Narinder Kaur, Surbhi Sharma, Anil K. Sharma, Deepak Kala, Hridayesh Prakash, Shagun Gupta

**Affiliations:** 1Department of Biotechnology, Maharishi Markandeshwar (Deemed to Be) University, Mullana, Ambala 134003, India; 2Department of Microbiology, Maharishi Markandeshwar (Deemed to Be) University, Mullana, Ambala 134003, India; 3NL-11 Centera Tetrahertz Laboratory, Institute of High Pressure Physics, Polish Academy of Sciences, 29/37 Sokolowska Street, 01142 Warsaw, Poland; 4Amity Center for Translational Research, Amity University Uttar Pradesh, Noida 201313, India

**Keywords:** cancer biomarkers, electrochemical biosensors, nanomaterials, cancer, cancer screening, transducers, point of care systems, biosensing platforms

## Abstract

Early and effective diagnosis of cancer is decisive for its proper management. In this context biomarker-based cancer diagnosis is budding as one of the promising ways for early detection, disease progression monitoring, and effective cancer therapy. Integration of Biosensing devices with different metallic/nonmetallic nanoparticles offers amplification and multiplexing capabilities for simultaneous detection of cancer biomarkers (CB’s). This study provides a comprehensive analysis of the most recent designs and fabrication methodologies designed for developing electrochemical biosensors (EB) for early detection of cancers. The role of biomarkers in cancer therapeutics is also discussed.

## 1. Introduction

Cancer is a cluster of diseases in which normal body cells unexpectedly begin to grow uncontrolled and develop first into Benign than malignant tumors. It is the leading cause of death across the globe and occupies first or second cause of mortality. Malignant tumors get metastasized to various locations and cause death of the individual. The major challenge in cancer treatment is the early-stage diagnosis of the disease which would improve the efficiency of cancer treatment and survival significantly. Currently, invasive techniques like imaging and biopsy-cytology are being used to detect cancers which are time consuming, cumbersome and costly in nature. In view of this, development of novel and cost-effective cancer detection strategies is paramount requirement.

A biomarker is any molecule, structure or activity that can be detected in the body and predicts the occurrence of any disease [1]. Cancer biomarkers can be a gene, gene product, etc. [2] which are released during genetic modification and cell division. Extensive studies have recognized distinctive forms of biomarkers that enable early detection of various type of cancer like lung, colorectal, pancreatic, breast, renal, pancreatic, and oral cancers [3]. Their presence is validated by examining human bodily fluids like urine, plasma, serum, blood and tumour cells. In view of its prospective with point of care (POC) demands, cancer biomarkers can be used in cancer prediction, and therapeutic monitoring [4].

For example, Prostate-specific antigen (PSA), is a biomarker used in the detection of prostate cancer [5]. Likewise, P53 can be used for the diagnosis of leukemia, lung, bone, breast, and ovarian cancers [6]. Furthermore, CA 15-3 (cancer antigen 15-3) is a potent biomarker for breast cancer detection [7]. In addition, CD44 is used for diagnosing breast cancer in different stem cells (BCSCs) [8]. Apo-A1 has emerged as a novel biomarker for diagnosing bladder cancer [9]. A recent study recognized Pseudopodium-rich atypical kinase one, SGK 26699 (PEAK1) as a potent pancreatic dental adenocarcinoma biomarker [10]. Also, different biomarkers like AFP, CA125, CEA and CA15-3 represent particular characteristics of four different lung cancer [11]. Sex-determining region Y-boX 2 (SOX2), another cancer biomarker is used for diagnosing various cancer types like prostate, skin, breast and lung cancer [12]. The emergence of these biomarkers enabled scientists to develop various techniques for instant and accurate diagnosis of cancer [13].

Currently, assays like high-performance liquid chromatography (HPLC), polymerase chain reaction (PCR), Western blotting, radioimmunoassay (RIA), enzyme-linked immunosorbent assay (ELISA), and immunohistochemistry (IHC), are employed for screening biomarkers. They do, however, have certain technological limitations in terms of their sensitivity, specificity, stability that restrict their use in resource restricted settings. A screening assay should be economical, point of use, non-tedious, fast and capable to operate in resource limited settings. Therefore, the development of novel, cost-effective ways of identifying cancer biomarkers is critical. During the last decade, many types of biosensors (Optical, electrochemical and mass sensitive) were reported for detecting cancer biomarkers in view of their remarkable properties like easy installation, higher sensitivity and specificity, low limit of detection, and multiplexing ability. The review here aims to give a comprehensive analysis of the most recent designs and fabrication methodologies specifically of EB to detect different cancer biomarkers. An overview of future prospects for the development of such effective EB’s is also presented.

## 2. Biosensors: An Analytical Approach for Biomarkers Detection

Biosensors have been extensively employed in biomarker detection as the most precise, rapid, and sensitive analytical assay. They detect cancer biomarkers by using specific biomolecules (antibodies, DNA, RNA, etc.) as biorecognition elements that are immobilized on the surface of the transducer [14], which converts the biological signals into detectable electrical or visual forms.

Based on the principle of transduction cancer biomarkers are categorized into several types like electrochemical, mass-sensitive, and optical. Biosensor designs differ according to the types of cancer biomarkers. For example, biorecognition molecules such as complementary nucleic acid probes, specific ligands, and specific antibodies are used to detect nucleic acid, receptors, and corresponding antigens, respectively.

## 3. Electrochemical Biosensors for Biomarker Detection

The biomarkers detection processes are mainly focused on the tracing of proteins on the membrane surface of tumor cells and/or cancer associated microRNA. Currently, various methods are reported for biomarker detection. However, electrochemical methods are usually preferred due to their sensitivity, specificity, affordability, the lower limit of detection and, the possibility of miniaturization [15,16,17]. EB’s are based on transducing the biological events into electrical signals generated after the electrochemical reactions with target molecules. A number of biorecognition elements like proteins, DNA, enzymes, aptamers etc. are being used to detect cancer biomarkers [18]. Biosensors can be categorized into immunosensors, aptasensors, enzymatic biosensors, and nucleic acid biosensors depending upon the biorecognition element being used. Here, we will focus on electrochemical methods that are employed for the detection of various cancer biomarkers.

### 3.1. Nucleic Acid-Based Biomarker Detection

Anti-tumor gene silencing, chromosomal degradation, and gene hypermethylation, are various cancer-causing defects that change the behavior of normal cells. These cancer-causing aberrations, including p53 gene mutation and micro RNAs (miRNA) are all categorized as nucleic acid-based cancer biomarkers. Nucleic acid-based biosensors are used for instant and accurate detection of cancer biomarkers due to their sensitivity in terms of detection of minute oligonucleotides concentration, single-base mismatch, and simple assembly [19].

These biomarkers allow for cancer detection even when individuals exhibit no physical symptoms. Wang et al. developed a magnetic-controllable EB [20] that enabled instant diagnosis of oral cancer biomarker, miRNA with a limit of detection (LOD) as low as 2.2 × 10^−19^ M. In addition, Boriachek et al. proposed an EB for detecting miRNA from human serum based on biotinylated complementary probes immobilization on magnetic beads (MBs) coated with streptavidin [21]. The electrochemical response was measured in terms of differential pulse voltammetry (DPV) with the LOD of 1.0 pmol/L. The proposed assay offers several advantages such as enhanced capture, low manufacturing cost, reduced assay time and matrix effect.

Furthermore, Luo et al. [22] developed a locked nucleic acid-based EB for detecting exosomal miRNA-21r with a LOD of 2.3 fM. The sensor was validated by electrochemical impedance spectroscopy (EIS) and DPV. A single-walled carbon nanotubes-based biosensor modified with fluorine-doped tin oxide was recently proposed for miRNA-21 detection [23], a specific biomarker for several types of cancer with LOD as low as 0.01 fmol L^−1^. The proposed biosensor revealed an acceptable performance in human serum and also good selectivity.

The strategies to develop the molecular biosensor for the detection of cancer biomarkers have been illustrated in Figure 1.

Hong et al. fabricated and developed an highly sensitive EB for detecting miRNA from human serum [24]. The proposed sensor was remarkably sensitive for miRNA-21 with a LOD of 100 aM. The self-assembled DNA concatemers can carry numerous RuHex that results in the significantly enhanced electrochemical signals. In another study, Topkaya et al. [25] fabricated a graphite-based biosensor to detect hypermethylation of the glutathione S-transferase P1 (GSTP1) gene, a marker for prostate cancer. The cancer biomarker was detected at a concentration of 2.92 pM as validated by DPV and EIS. Compared to biopsy, it was a simple procedure and might be used as a diagnostic tool for patients susceptible to prostate cancer. Peng et al. [26] fabricated an immunosensor for detecting multidrug resistance (MDR) gene using Au nanoparticles/toluidine blue–graphene oxide modified electrode. The electrochemical response was studied in the form of EIS and DPV with a detection limit of 2.95 × 10^−12^ M. Azmi et al. [27] proposed that a silicon-based EB can accurately identify prostate cancer by using 8-hydroxydeoxyguanosine as a biomarker with a LOD as low as 1 ng/mL. They claimed that the sensor is extremely quick and may be regarded as a possible option for point of care application in terms of easy assembly and ultra-sensitivity. Chen et al. created a molecular biosensor for detecting Cytokeratin 19 fragment 21–1, a biomarker for non-small cell lung cancer detection using functionalized three-dimensional graphene and silver nanoparticles [28]. The thiol labelled probe DNA was utilized for the detection of its complementary target DNA. A strong signal and peak current were shown by CYFRA21-1 using a linear relationship in different dilutions ranging from 1.0 × 10^−14^ to 1.0 × 10^−7^ M, confirming the presence of LC. Shoja et al. proposed another biosensing platform to recognize the EGFR exon2 point mutation L858R based on rGO/functionalized nanoparticles on PGE surface [29]. The use of nanoparticles for the immobilization of the ssDNA capture probe resulted in higher sensitivity (0.0188 mA/M) with a detection limit of 120 nm.

Li et al. created a genosensor to know specific sequences deduced from the maternally expressed gene3 (MEG3). The sensor was based on a nanocomposite interface of graphene-like tungsten disulfide/dendritic gold nanostructures [30]. The lower detection limit for both the sequences was found to be 0.25 fM and 0.3 fM respectively. A number of DNA-based biosensors have been discovered to detect biomarkers specific to breast cancer. Zhao et al. proposed a molecular aptasensor for detecting carbohydrate antigen 15-3 (CA15-3), a breast cancer biomarker [31]. DNA tagged with MoS2 nanosheets (NSs) was employed as a detection probe for measuring CA15-3. The constructed biosensor performed well, with an enhanced LOD of 0.0039 U/mL. Another study enabled the detection of BRCA1 gene by utilizing AuNPs-based DNA hybridization biosensor [32]. Polyethylene glycol/AuNPs composite was utilized for the immobilization of the DNA probe on the glassy carbon electrode (GCE) surface, and EIS was used to study the detection performance. Cui et al. have created a tagged free EB for detecting BRCA1 based on low-fouling zwitterionic peptide self-assembled monolayer (SAM) support, 19-mer gene related sequence-specific oligonucleotides, and EIS as a detection technique [33]. The LOD of the sensor was determined to be 0.3 fM. Another team of researchers proposed a ferrocene-cored poly (amidoamine) based molecular biosensor for detecting BRAC1 gene [34] with a LOD of 0.38 nM. Additionally, Saeed et al. fabricated an EB for detection of breast cancer biomarkers (ERBB2 and CD24). The biosensor strategy is based on a sandwich assay by immobilizing the capture probe of the target molecule onto graphene oxide-modified gold nanoparticles [35]. The LOD for ERBB2 and CD24 was found to be 0.16 nM and 0.23 nM, respectively. Topkaya et al. developed a molecular sensor-based EB for PSA detection. In this study, methylation and unmethylated GSTP1 sequences were immobilized on the fabricated electrode surface, and DNA hybridization study was assessed using electrochemical techniques. The LOD of this biosensor was determined to be 2.92 pM. Fayazfar et al. created an impedimetric biosensor for detecting tumour protein 53 (TP53) gene mutations. The fabrication approach enabled gold nanoparticles (AuNPs) to grow on MWCNTs and capture the cDNA sequence over DNA probe. The LOD of the sensor was found to be 1.0 × 10^−17^ M [36]. Cao et al. designed a multiple electrochemical aptasensor based on covalent-organic framework that incorporated AuNPs labeled with Ag and Cu_2_O nano-clusters for sensitive and simultaneous detection of mi-RNA 155 and mi-RNA 122. It was observed to have wider linear range of 0.01–1000 pM and ultrasensitive LOD of 6.7 and 1.5 fM, respectively [37].

### 3.2. Protein/Immuno Based Biomarker Detection

Protein biomarkers have gained prominence due to the technological advancements in analytical instruments for detection and quantification of proteins in complicated biological materials [38]. Different studies have been carried out to develop biosensors that enable the detection of protein-based cancer biomarker in biological samples (Figure 2). Elshafey and his colleagues proposed a sensor for detecting epidermal growth factor receptor (EGFR) in human plasma and phosphate buffer with detection limits of 0.88 pg/mL and 0.34 pg/mL, respectively [39]. The working surface of the gold electrode was enhanced by deposition of AuNPs onto which G protein was immobilized and subsequently EGFR was detected using EIS. Ilkhani et al. also fabricated an aptamer based biosensing assay by attaching biotinylated EGFR aptamer for detection of EGFR. The LOD of the proposed sensor was 50 pg/mL [40].

Zhang et al. developed a nanocomposite-based electrosensing platform for carcinoembryonic antigen (CEA) detection by immobilising anti-CEA antibody [41]. The sensor verified the detection of CEA, with an excellent detection limit of 1 × 10^−4^ ng mL^−1^. The biosensor’s performance was investigated using CV, EIS and linear sweep LSV. Luo et al. proposed another immunosensing platform based on SWCNTs, quantum dots and reduced graphene oxide-AuNPs to detect CEA [42]. The interface properties of modified electrodes were investigated using square wave voltammetry (SWV), CV and EIS. The detection limit of 5.3 pg/mL was recorded.

In another study [43], human epidermal growth factor receptor 2 (+) BT 474, a well-known breast cancer biomarker, was detected utilizing the HER 2 antibody with a LOD of 4.7 × 105 exosomes/μL. Canbaz and his colleagues covalently bonded the complementary HER3 Ab on a nanomodified gold electrode to detect another cancer biomarker called HER3 [44]. Voltammetry and EIS were used to characterize the reported sensors, which revealed a linear detection range of 0.2–1.4 pg/mL. In another study, a disposable EB was used to detect the breast cancer biomarker, HER2-ECD in human blood with a LOD of 2.1 ng/mL. The sensitivity of the reported sensor was confirmed by screening several human proteins as well as another cancer biomarker viz. CA15-3 [45]. Furthermore, Yang et al. developed an AuNP’s modified electrode based nano hybrid sensor to detect HER2 in human blood using DPV. The LOD was as low as 4.9 ng/mL [46]. Likewise, Pacheco et al. revealed a polymer-based EB to recognize HER2-ECD, the breast cancer biomarker with a LOD of 1.6 ng/mL [47]. Furthermore, Freitas et al. developed an immuno based sensor by using LSV based technique on carboxylic acid-activated MBs to detect HER2 in human serum with the LOD of 2.8 ng/mL [48]. Furthermore, Carvajal et al. proposed a disposable inject-printed EB capable of detecting HER2, in a concentration of 12 pg/mL within 15 min [49]. Another group demonstrated HER2 detection utilizing a cerium oxide-monoclonal Ab-based immunosensor with the detection limit of 34.9 pg/mL [50]. A modified gold electrode with magnetic GO was developed by Lin and his colleagues to detect vascular endothelial growth factor (VEGF) by immobilizing Avastin as a bio-recognition element [51]. The suggested biosensor was tested in human plasma at concentrations ranging from 31 to 2000 pg/mL. Furthermore, an amperometric immunosensor was developed for detecting VEGF with a LOD of 38 pg/mL [52]. To diagnose target biomarkers, a disc-shaped carbon fiber micro-electrode device was developed to covalently link ferrocene monocarboxylic acid tagged anti-VEGF antibody. Pang et al. [53] discovered that extracellular vehicles (EVs), including exosomes, convey molecular information and are used as cancer biomarkers. Rabbit anti-human CD9 antibody was also used to modify Au electrode to identify exosomal surface protein from human serum [54] using a sandwich electrochemical immunosensor. Horseradish per-oxidase (HRP)-conjugated anti-IgG antibodies were used as the bio-recognition element. The immunosensor demonstrated detection sensitivity of 200 exosomes/L.

In addition, Jeong and his group proposed a sensor for detecting multiple exosomal proteins with a detection limit of 3 × 10^4^ exosomes from plasma [55], fabricated nano-tetrahedron (NTH)-assisted aptasensors for detection of exosomes from liver cancer cells (Figure 3), with LOD of 2.09 × 10^4^ exosomes/mL. During the experiment, NTH factionalized Au electrode was prepared, to which LZH8, a specific aptamer was immobilized that was capable of detecting target exosomes.

Ho et al. proposed a disposable EB for detecting alpha-enolase 1 (ENO1), a specific lung cancer biomarker, with a LOD of 11.9 fg/mL [56]. Huang and his team discovered an immunochemical aptasensor for detecting exosomes originating from gastric cancer. Anti-CD63 mediated exosome capture probe was used to capture human plasma exosomes, with only gastric cancer exosomes interacting with rolling circle amplification (RCA), producing in the G-quadruplex. After incubation, a heme-G-quadruplex structure was formed, which displayed electrochemical response indicating detection of a gastric cancer biomarker with a LOD of 9.54 × 102 exosomes/mL [57].

In 2019, Heiat et al. used a modified Au electrode with spindle-shaped gold nanoparticles to develop an electrochemical aptasensor to detect alpha fetoprotein (AFP). The LOD was observed to be 0.23 pg mL^−1^, with a linear range of 0.005 to 10 ng mL^−1^ [58]. Another work used bio-synthesized (by processing biomass, floral, and peanut cells) nitrogen-doped mesoporous carbon nanostructures as a label-free electrochemical aptasensor to detect AFP. With a LOD of 61.8 fg mL^−1^, the linear detection range was found to be 0.1 pg mL^−1^ to 100 ng mL^−1^ [59].

A sensitive electrochemical aptasensor was created using coral-like poly-aniline (PANI)/AuNPs and peptides for the detection of PSA. The aptasensor has a LOD of 0.085 pg mL^−1^ and a high sensitivity of 462.7 μA (ng mL^−1^)^−1^ cm^−2^ [60]. A bi-functional electrochemical aptasensor was designed by coating it with bimetallic NiCO Prussian blue analogue (NiCoPBA) nanocubes for the detection of CEA. It was observed that it has extremely low LOD of 0.74 fg mL^−1^ (1.62 fM) with the linear range of 1.0 fg mL^−1^ to 5.0 ng mL^−1^, as well as an LOD of 47 cells mL^−1^ for H460 cells [61]. A high-performance electrochemical aptasensor was developed for the detection of CA 125 by fabricating the amidoxime-modified polyacrylonitrile nanofibers doped with Ag nanoparticles (AgNPs-PAN-oxime NFs) on the surface for better conductance [62]. It was resulted that such sensor has dynamic linear range (DLR) from 0.01 to 350 U mL^−1^ with a correlation value of 0.991 with LOD of 0.0042 U mL^−1^. Gu et al. developed an aptasensor assembly for the immobilization of bio-recognition element HER2 and MCF-7 cells using bi-metallic ZrHf-MOF framework with enumerous carbon dots (CDs) [63]. The sensor developed was capable to detect HER2 in a dynamic range from 0.001 to10 ng mL^−1^ with the LOD of 19 fg mL^−1^ whereas MCF-7 cells were linearly ranged from 1 × 10^2^ to 1 × 10^5^ cell mL^−1^ with a rough LOD of 23 cell mL^−1^.

Moon et al. created an immunosensor to detect PSA by immobilising specific antibody on Au electrode [64,65]. Au nanowire (NW) was coated with polypyrrole to improve the antibody immobilization. The LOD was found to be 0.3 fg/mL. A similar study was carried out by Yang et al. [65] who developed quantum dots (QDs) modified graphene oxide (GO)-based EI to detect PSA in human serum samples. Graphene sheets (GS) were employed as vehicles for primary anti-PSA Ab and QD-functionalized GS as carriers for secondary anti-PSA Ab. The immunosensor showed a LOD of 3 pg/mL. Heidari et al. [66] presented a GCE modified sensor for detecting and quantifying p53 cancer biomarkers. CdS nanocrystals (CdS NCs) were immobilized on GCE and AuNPs were added to the process through formation of a sandwich-type immune complex between first anti-p53/p53/secondary anti-p53. A LOD of 4 fg/mL for p53 was deduced from the developed sensor.

Lin et al. [67] used similar platforms for detecting alpha-methylacyl-CoA racemase, another putative prostate cancer biomarker. Rauf et al. proposed a GO based SPCE modified with carboxylic acid [68] for detecting Mucin1 (MUC1), a cancer biomarker in human serum with a LOD of 0.04 U/mL. Bravo et al. proposed an immunosensor based on AgNPs coated with polyvinyl alcohol for the detection of epithelial cell adhesion molecule, an epithelial cancer biomarker. The authors demonstrated that the suggested immunosensor may be used in medical applications with LOD as low as 0.8 pg/L [69]. Zhao et al. developed a new technique for detecting cancer stem cell biomarkers (CSCs), CD44 for breast cancer, with LOD’s of 2.17 pg/mL and 8 cells/mL, for CD44 and CD44-positive CSC respectively [70]. Furthermore, Zeng et al. created an EI paired with a signal amplification method to detect CYFRA21-1, a biomarker for NSCLC, at a concentration of 43 pgmL^−1^ [71]. The suggested immuno hybrid sensor was created by combining chitosan, graphene nanoparticles, and glutaraldehyde on GCE surface, which gives superior conductivity. The developed sensor was claimed to have outstanding analytical performance.

Likewise, Aydn and Sezgintürk developed an ITO-based EI for detecting NSE, a cancer biomarker specific for a variety of malignancies such as lung adenocarcinoma, squamous cell carcinoma, skin cancer, breast cancer, prostate cancer etc. with a LOD of 6.1 fg/mL (Figure 4) [72].

Mattarozzi et al. detected human epididymis protein 4 (HE4), a biomarker specific to epithelial ovarian cancer, in human blood at concentrations as low as 2.8 pM [73]. Pacheco et al. developed a gold screen printed based biosensor for the detection of CA 15-3 in human serum with LOD as low as 1.5 U/mL [4]. Kim et al. developed another EI for detecting apolipoprotein-A1 proteins, an early bladder cancer biomarker. The sensor was developed using electrochemical ELISA on an ITO electrode with LOD of 1 pM [9]. Rajaji et al. [74] proposed an iron nitride NPs fabricated reduced GO sheets-based biosensor to detect 4-nitroquinoline N-oxide (4-NQO), with a LOD of 9.24 nM. In addition, Prasad et al. fabricated a paper electrode modified with GO-based EB to detect SGK269, a pancreatic cancer biomarker. The sensor showed a detection limit of 10 pg/mL. Martin et al. [75] recently proposed a MBs-based micro fluidic EB to detect hypoxia-inducible factor-1 alpha (HIF-1), a tumoral hypoxia biomarker with LOD as low as 76 pg/mL. In another paper, Mathew and his colleagues [76] developed an EI using nano-finger electrodes to detect prostate tumor having extracellular vesicles with a LOD of 5 tdEVs/mL. Similarly, Munge and his team [77] developed a SWCNT-based EI to detect matrix metalloproteinase-3 (MMP-3), by introducing HRP(14-16) for antibody conjugation, and polymer bead loaded multi-enzyme. Their response was studied in terms of LOD’s of 0.4 ng/mL (7.7 pM) and 4 pg/mL (77 fM), respectively. Table 1 summarizes EB’s and their performance in detecting cancer biomarkers.

### 3.3. Electrochemical Aptasensors for Cancer Biomarker Detection

Now a day’s aptasensors are commonly used in research for the detection of cancer biomarkers. Electrochemical aptasensors for cancer biomarkers can be divided into three groups including: (1) those detecting protein tumor biomarkers such as PSA, carcinoembryonic antigen (CEA), and mucin 1 (MUC1); (2) electrochemical aptasensors for detecting circulating tumor cells (CTCs), such as EpCAM; and (3) electrochemical aptasensors for exosomes (i.e., oncoproteins, RNA, and DNA fragments) such as CD63 [80]. An electrochemical aptasensor based on microgel nanocomposite was developed for the highly sensitive detection of miRNA-21 as a biomarker. In this study, the microgel particles of AuNPs were wrapped in a mixture of acrylic acid (AAc) and N-isopropylacrylamide (NIPAm) for polymerization. Then, the gold electrode surface was modified with this porous network structure of AuNP@NIPAm–co–AAc microgels [81]. The amino-modified DNA capture probe (complementary to the target RNA) was bound to the activated carboxyl groups of the particulate gel on the surface of the electrode and applied for miRNA-21 detection using DPV. The assay showed a LOD of 1.35 aM with a linear range of 10 aM to 1 pM. A sandwich-type electrochemical aptasensor was developed for the simultaneous detection of carcinoembryonic antigen (CEA) and cancer antigen 15-3 (CA 15-3). A nanocomposite of AuNPs and 3D graphene hydrogel (AuNP/3DGH) was employed as a biosensing substrate for the surface modifification of a glassy carbon electrode (GCE). The AuNP/3DGH-modifified GCE was treated with 3-mercaptopropionic acid (MPA) in order to provide carboxylic groups and immobilization of 50 -amino-functionalized CEA aptamer I (CEAApI) and CA 15-3 aptamer (CAAp) on the surface of the electrode. The LODs of CEA and CA 15-3 were found to be 11.2 pg mL^−1^ and 11.2 × 10^−2^ U mL^−1^, respectively. The aptasensor was used for the simultaneous detection of biomarkers in the serum samples [82].

An electrochemical aptasensor using LC-18 aptamer was developed for the detection of lung cancer biomarkers in real blood plasma [83]. The LC-18 aptamer is a highly specifific aptamer with the ability to identify lung cancer-related proteins and cells. The aptasensor was fabricated using the immobilization of thiolated aptamer on the surfaces of gold disc electrodes. In order to inhibit any non-specifific interactions, the uncovered surfaces of modifified electrodes were blocked using blocking thiolated oligonucleotides. CV and non-Faradic EIS were used for the characterization of the aptasensor and when analyzing samples to detect analyte.

A label-free electrochemical aptasensor using a nickel hexacyanoferrate (NiHCF) nanocube as the in-situ signal probe and polydopamine-functionalized graphene (PDA@Gr) as the substrate was developed for the detection of CA-125 [84], a marker for diagnosis of ovarian cancer. The aptasensor exhibited a low LOD of 0.076 pg mL^−1^ with a linear range of 0.10 pg mL^−1^ to 1.0 µg mL^−1^.

A glassy carbon electrode employed with GO and a dendritic gold nanostructure (DenAu) was used as the transducer for determination of cancer exosomes [85]. A thiolated CD63 aptamer was immobilized to the surface of glassy carbon electrode (Au-S interaction). Alkynyl-4-ONE’s aldehyde groups were then covalently attached to the amine groups of exosomal proteins, causing alkynyl-4-ONE to be fixed on exosomes (Figure 5). The electrode’s surface was then sprayed with N3-DNA, which reacted with alkynyl-4-ONE. Thereafter, electrode’s surface was subsequently immobilized with H1 and H2 biotinylated aptamers to establish a hybridization chain reaction (HCR) and then coated with SA-HRP. Thus, formed HRP oxidized H_2_O_2_ to H_2_O and O-phenylenediamine (OPD) to 2, 3-diaminophenazine (DAP). This aptasensor was capable to detect cancer exosomes in the range of 1.12 × 10^2^ to 1.12 × 10^8^ particles µL^−1^ with a LOD of 96 particles µL^−1^.

An aptamer-based sensor was again used by other research group for the diagnosis of MDA-MB-231 breast cancer cells [86]. Here, non-spherical gold NPs were electrodeposited on the surface of an Au electrode to use it as a transducer for the immobilization of aptamer. A thiolated aptamer (83-mer) showing high affinity against MDA-MB231 breast cancer cells was anchored on the surface of a gold nanostructure through covalent bonding between Au-S for its detection. It was observed that the developed aptasensor was successful to detect MDA-MB-231 breast cancer cells at concentrations ranging from 10 to 1 × 10^3^ cell mL^−1^, and LOD was roughly taken as 5 cells mL^−1^.

### 3.4. Biosensors for Diagnosing Multiple Biomarkers

There is no single biomarker for specific malignany, so there has been a rise in research interest in establishing a multi-biomarker platform, as evidenced by a few literature surveys. Chen et al. [87] created a sandwich electrochemical platform based on biofunctional carboxyl graphene nanosheets (CGS) by sequentially immobilising anti-CEA and anti-AFP antibodies to detect CEA and alpha-fetoprotein (AFP) with a limit of detection of 0.1 ng/mL and 0.05 ng/mL respectively. Atlintas and his colleagues [88] produced a biosensor capable of tracing CEA and EGFR over a linear range of 20–1000 pg/mL. CA15-3 was also detected over a wide range of 10–200 U/mL. The signal amplification was enhanced by the deposition of AuNPs layer on the surface of electrode followed by the immobilization of specific antibodies to detect CEA and EGFR. Chikkaveeraiah et al. [89] reported an EI capable of detecting two prostate cancer biomarkers, PSA and IL-6, with detection limits of 0.23 pg/mL and 0.30 pg/mL, respectively. Wan et al. discovered a comparable biosensor based on modification of SPCE with MWNT [90]. The sensor was capable of detecting prostate cancer biomarkers, PSA and IL-8, with detection limits as low as 5 pg/mL and 8 pg/mL, respectively. Hong and colleagues [91] described an Au modified ITO electrode capable of detecting CEA, CA125 and PSF, with LOD of 0.7 pg/mL, 0.007 U/mL and 0.9 pg/mL, respectively. Similarly, a biotin-doped polypyrrole EI was developed for detecting CEA, CA125 and PSF tumour biomarkers with LODs of 0.8 pg/mL, 0.005 U/mL and 0.7 pg/mL, respectively [92]. Wang et al. developed a paper-based aptasensor for simultaneous detection of cancer biomarkers, NSE and CEA, with a LOD of 10 pg/mL and 2 pg/mL respectively [93]. In addition, Tang et al. proposed a biosensor for detecting four distinct lung cancer biomarkers, namely AFP, CEA, CA125, and CA15-3, with a LOD of <0.5 g/L from susceptible patients’ blood or urine. Furthermore, Wilson and Nil established another biosensor for detecting seven cancer biomarkers namely CEA, hCG, AFP, CA125, CA15-3, ferritin, and CA19-9, that are linked to various forms of cancer [94].

## 4. Biomarkers in Cancer Therapeutics

The detection of cancer biomarkers can be of great importance in personalized medicine. Even with malignancies of the same type, each person’s biomarker pattern is different and influences the choice of cancer treatment strategy. The identification of biomarkers is essential for identifying abnormalities and selecting treatment options. Cancer biomarkers are important therapy targets. The biomarkers can even help in monitoring the effect of cancer treatment on patients.

The cancer biomarkers-specific monoclonal antibodies conjugated with the radiolabeled isotopes, drugs, and bispecific T cell engagers are utilized in immunotherapy-based approaches to target the cancer cells [95,96,97,98]. CD-19, CD-20, CD30, CD33, and some other biomarkers have been targeted so far for cancer immunotherapies. Among all of these, CD-19 is the mostly targeted biomarker, and a number of products, including blinatumomab and CAR-transduced T cells (CAR-T), have already been approved for use in clinical trials for multiple myeloma [99]. The other widely used cancer biomarker targeted for the treatment of lymphoid malignancy is CD 20 [100]. 

A new strategy for specific cancer immunotherapy centered on immunological checkpoints. T-cell immunological activity is negatively regulated by immune checkpoints called programmed death 1 (PD-1) and cytotoxic T-lymphocyte-associated antigen 4 (CTLA-4). New immunotherapies for melanoma, non-small cell lung cancer, and other cancers have been developed as a result of the inhibition of these targets, which boosted immune system activation [101]. The immune checkpoint inhibitors (ICIs) targeted the immune system rather than the cancer cells to restore the suppressed cancer immunity. The development of targeted medicines is being driven by more immune checkpoint biomarkers, including IDO, LAG3, TIM-3, TIGIT, SIGLECs, VISTA, and CD47.

The extracellular vesicles mediated therapeutic approach is also emerging as a promising tool for cancer treatment [102]. Exosomes and microvesicles are two major types of EVs that have been recognized so far. The extracellular vesicles (EVs) consists of protein, DNA, mRNA, micro-RNA and is involved in intercellular communication. Both healthy and cancerous cells release EVs, that regulate other cells at the primary tumour sites and in distant tissues. The EVs secreted by cancer cells establish lasticity, immunomodulation, resistance to the therapy and pre-metastatic niche. So, targeting the cancer derived EVs is emerging as a new therapy for cancer treatment and diagnosis. Another application of EVs is as a delivery vehicle for anticancer drugs and vaccines. The EVs can be modified for developing tissue specific targeted drug delivery system. The cancer specific biomarkers can be targeted by aptamer or mABs modified EVs for accurate and effective cancer treatment.

The next generation of developments in cancer therapy is being driven by the search for novel biomarkers, new designs, and delivery strategies like nanotechnology of targeted drugs.

## 5. Conclusions and Future Perspective

In order to enhance cancer prognosis and therapy, there is a high demand for low-cost biosensors that can detect crucial biomarkers quickly and precisely. This review was aimed to give a comprehensive analysis of electrochemical biosensors involved in detection of different cancer biomarkers. However, technologies like nano-fabrication and clinical applications are required to provide cost-effective and unique biosensing devices. Surprisingly, EB’s are the most often mentioned platform for cancer biomarkers detection. Nanostructured materials and nanocomposites are the key components in the manufacturing and design of different EB’s, and they are expected to be employed more in future research. After reading all the attempt of different researcher some points could be considered for its economical and user-friendly device development. (1) Researchers may be able to better control the shape and size of nanostructures by using procedures and chemicals to synthesis less hazardous (ideally biogenesis) nanostructures and nanocomposites. (2) Using cutting-edge, low-cost, and more accurate aptamer selection, synthesis, and functionalization methods, the signal transducer’s aptamer stabilization approach should also be fine-tuned so that the immobilization procedure can be carried out with high repeatability and entire dependability. (3) It’s now the time to develop cancer apta-sensors into commercially viable electrical chips to use in hospitals. This can be achieved through the designing of signal transducers in such a way that it can immobilize multiple aptamers at a time for different diseases. We feel that developing sensor would not only facilitate early diagnosis but also enable us to manage cancer more effectively.

## Figures and Tables

**Figure 1 vaccines-10-02138-f001:**
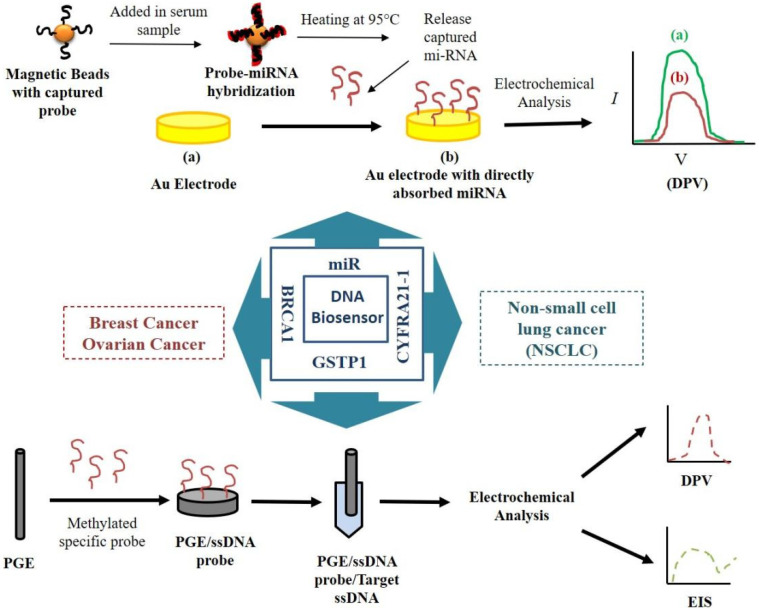
Schematic diagram of nucleic acid-based biosensors for detecting cancer biomarkers.

**Figure 2 vaccines-10-02138-f002:**
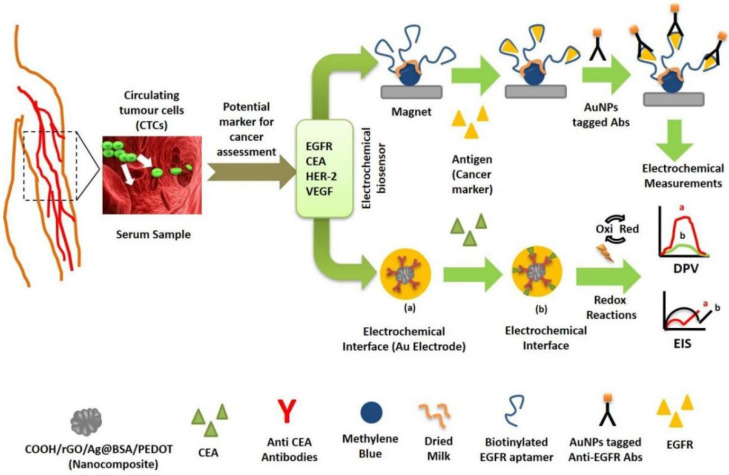
Illustration of EBs based on detection of cancer marker proteins.

**Figure 3 vaccines-10-02138-f003:**
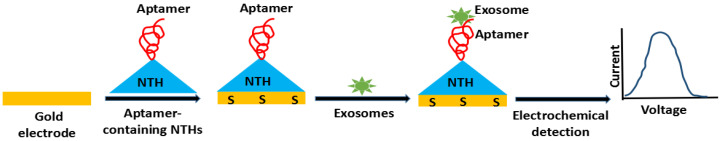
Schematic illustration of DNA nanotetrahedron (NTH)-assisted aptasensor.

**Figure 4 vaccines-10-02138-f004:**
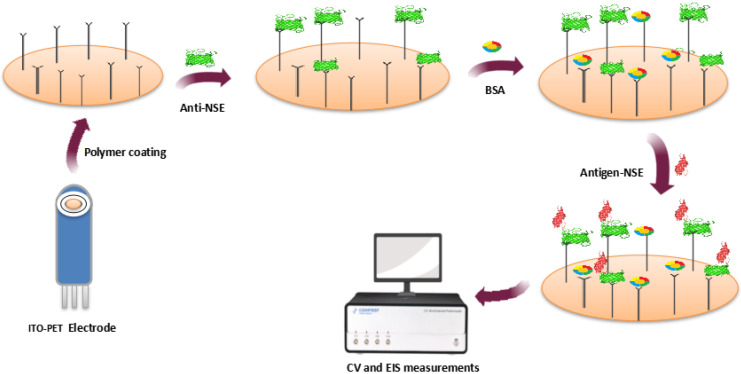
Immunosensor for NSE determination.

**Figure 5 vaccines-10-02138-f005:**
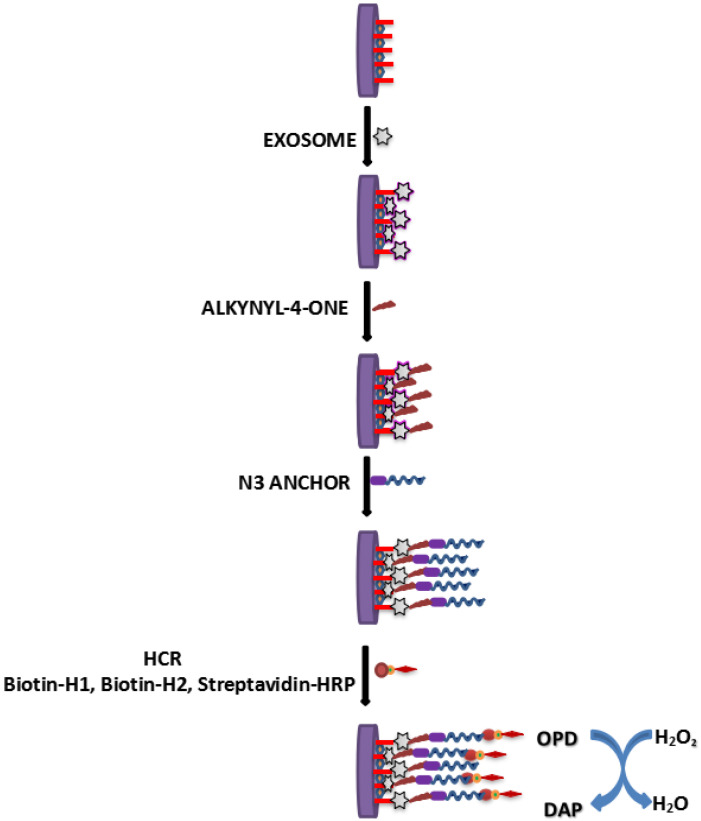
Schematics determination of cancer exosomes using aptasensor.

**Table 1 vaccines-10-02138-t001:** Electrochemical biosensing platforms to detect cancer biomarkers.

Biosensing Platform	Functionalization Method	Analyte	LOD	Reference
AuNP modified Au electrode	Cysteamine/PDITC/protein G	EGFR	0.34 pg/mL	[39]
Antibody-functionalized	Silicon nanowire	8-OHdG	1 ng/mL	[27]
Immunosensor based on direct incorporation of anti-PSA	Polypyrrole (Ppy) nanowire	PSA	0.3 fg/mL	[64]
DNA biosensor	Gold nanoparticles/toluidine blue–graphene oxide nanocomposites	MDR1	2.95 × 10^−12^ M	[26]
Streptavidin-coated magnetic beads (MB) based sensor	Gold nanoparticles	EGFR	50 pg/mL	[40]
Polyethylene glycols	Gold nanoparticles	BRCA1	1.72 fM	[32]
GCE	Gold	CEA, CCA, CA125, CYFRA21-1,NSE	0.2 ng/mL, 0.03 ng/mL, 0.9 U/mL, 0.4 ng/mL, 0.9 ng/mL	[78]
Glassy carbon electrode	Reduced graphene oxide/Au (rGO/Au)	CEA, A199,CA724, AFP	8.1 pg/mL, 0.0076 U/mL, 0.0069 U/mL, 6.3 pg/mL	[79]
Indium tin oxide (ITO) based electrode	Gold nanoparticles-modified graphene oxide	SOX2	7 fg/mL	[12]
GCE	Gold nanoparticles and graphene oxide	HER2	0.16 nM	[35]
Three-dimensional graphene based	Ag nanoparticles	CYFRA21-1	1.0 × 10^−14^ M	[28]
Mesoporous carbon/Ni-oxytetracycline metallopolymer nanoparticles modified pencil graphite electrode	Reduced graphene oxide	EGFR exon 21	120 nM	[29]
Composite interface	Carboxylated single-walled carbon nanotubes	MEG3	0.25 fM	[30]
Immunosensor	CdS nanocrystals/graphene oxide-AuNPs based electrochemiluminescence	p53	4 fg/mL	[66]
Au electrode surface	Ferrocene-cored poly(amidoamine) dendrimers	BRCA1	0.38 nM	[34]
SWCNT-grafted	Dendritic Au nanostructure	miR-21	0.01 fM/L	[23]
DNA based aptameric biosensor	Interface interaction of MoS2 nanosheets	CA15-3	0.0039 U/mL	[31]

## Data Availability

Not applicable.

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
