# Peer review of "Current Update on Biomarkers for Detection of Cancer: Comprehensive Analysis"

_vaccines, 2022, doi:10.3390/vaccines10122138_

Round 1

Reviewer 1 Report

The review aims to give an overview on biosensors (mainly electrochemical) for detection of cancer biomarkers.

General comments:

1. The title says “emphasizing therapy benefits”, however I didn’t notice much of this emphasis in the paper to be honest.

2. I do not fully understand why the authors chose “Vaccines” journal and not a more related journal such as Biosensors, Biomolecules, Sensors etc.

3. Line 118: Is figure 2 (and all other figures) totally drawn by authors or taken from other papers? If from other sources need to get a permission.

4. For me stating the functionalization method and reached LOD for each of the paper is not enough. What was the significance of the study (is this the first ever biosensor? Does it have far lower LOD? Was it the only one used for clinical sample testing? Was the platform cheaper than the others? how does it stand in comparison to other biosensors employed to detect similar analyte of interest? or something in this term should be added to make the paper more appealing. At least for some papers this should be added.

5. Table 1. Can split the first column into two sections: Biosensing platform and functionalization method to make it more readable I think.

There are some errors/misunderstandings of the terms used and issues in organization of the paper:

1. Lines 24-25: The first sentence is incorrect because leukemia is a type of cancer formed in the blood but not the term that describes tumor mass in general.

2. Lines 32-33: Biosensor could be defined in a better way and not as a “representative of disease”. The presence or absence (or increased or lowered value) of biomarker tells us if one has a disease or not.

3. Line 67: cancer cells are usually used as analytes and not as biorecognition elements as well as I know

4. Line 72-73: in Figure 1 the way words under the arrows (aptamer, cancer cell, antigen etc.) are used is not very clear. What are they: biorecognition elements or analytes? Biomolecules that is mentioned for example encompasses almost everything mentioned there. Antigen is also a very broad term.

5. Lines 85-86: First sentence is not correct. Not all biomarkers are present on cancer cell surfaces. What do the authors want to say here?

6. Line 100: PNA and Aptamers are not biomarkers, they are used as biorecognition elements.

7. Line 171: the title of the section should be renamed to “Detection of protein biomarkers”. Protein—based means the biorecognition element is a protein.

8. Line 185 and all over: CEA detection is based on antibodies but is in “protein” section. The overall classification of the paper is not very clear. The paper should be classified by in only one of the ways I think:

By biorecognition element: antibody, enzyme, nucleic acid probes (for hybridization), aptamers (both nucleic acid-based and peptide-based)

By the biomarker nature being detected: ions, proteins, nucleic acid (genes, RNAs), antibodies (autoantibodies), cells (microbial, cancer etc.)

By electrochemical biosensor types (DPV, EIS etc)

Right now the classification is messy. 3.1. discusses biosensors where DNA was used as a biorecognition element.

3.2 (Protein) discusses detection of

- protein biomarkers

- exosomes which are not proteins but are more complex structures

- cells!

For most of these analytes biorecognition elements are antibodies and sometimes aptamers. So by “protein” do the authors mean analyte (then why discuss cells and exosomes?) or biorecognition element (then why discuss aptamers?)

3.3. discusses detection of proteins by biosensors with antibodies as biorecognition elements.

So what is the difference between sections 3.2. where antibodies detected proteins and 3.3. is not clear to me.

9. Line 356: the word “different” can be termed as “multiple” or “more than one” perhaps.

Abbreviation of terms and typos:

1. This has to be done in the very first mentioning of the term

2. This should not be repeated multiple times throughout the paper as it is done for instance for PSA.

3. Some terms used only once should not be abbreviated

4. CB – better to leave this as cancer biomarker without abbreviation (this is my personal opinion)

5. Consistency of use should be kept. In one place Apt stands for aptamer while in others it is used simply as aptamers.

6. Please correct H2O and H2O2

Need more extension in the following:

1. Lines 52-55: What are the actual limitations of the assays?

Author Response

The review aims to give an overview on biosensors (mainly electrochemical) for detection of cancer biomarkers.

General comments:

  1. The title says “emphasizing therapy benefits”, however I didn’t notice much of this emphasis in the paper to be honest.

Response: The title has been modified to “Current update on Biomarkers for Early detection of Cancer: Comprehensive analysis”.

  1. I do not fully understand why the authors chose “Vaccines” journal and not a more related journal such as Biosensors, Biomolecules, Sensors etc.

Response:

We planned this paper for the special issue which deals with the cancer treatment where early detection is decisive for the treatment. So, from the technical point of view, the content of paper is deemed fit to the special issues.

  1. Line 118: Is figure 2 (and all other figures) totally drawn by authors or taken from other papers? If from other sources need to get permission.

Response: The figures were drawn by the authors.

  1. For me stating the functionalization method and reached LOD for each of the paper is not enough. What was the significance of the study (is this the first ever biosensor? Does it have far lower LOD? Was it the only one used for clinical sample testing? Was the platform cheaper than the others? how does it stand in comparison to other biosensors employed to detect similar analyte of interest? or something in this term should be added to make the paper more appealing. At least for some papers this should be added.

Response: As suggested the content has been added and marked in yellow.

  1. Table 1. Can split the first column into two sections: Biosensing platform and functionalization method to make it more readable I think

Response: Necessary changes have been incorporated in Table 1. 

There are some errors/misunderstandings of the terms used and issues in organization of the paper:

  1. Lines 24-25: The first sentence is incorrect because leukemia is a type of cancer formed in the blood but not the term that describes tumor mass in general.

Response: The sentence has been modified and marked in yellow.

  1. Lines 32-33: Biosensor could be defined in a better way and not as a “representative of disease”. The presence or absence (or increased or lowered value) of biomarker tells us if one has a disease or not.

Response: The sentence has been modified and marked in yellow.

  1. Line 67: cancer cells are usually used as analytes and not as biorecognition elements as well as I know

Response: The term cancer cells has been removed as biorecognition element.

  1. Line 72-73: in Figure 1 the way words under the arrows (aptamer, cancer cell, antigen etc.) are used is not very clear. What are they: biorecognition elements or analytes? Biomolecules that is mentioned for example encompasses almost everything mentioned there. Antigen is also a very broad term.

Response: Figure 1 is the Schematic representation of the working principle of biosensors for the detection of cancer biomarkers. For clarity the terms biomolecules and antigen are removed from the figure.

  1. Lines 85-86: First sentence is not correct. Not all biomarkers are present on cancer cell surfaces. What do the authors want to say here?

Response: The authors meant to say that the biomarkers detection processes are mainly focused on the tracing of proteins on the membrane surface of tumor cells and/or cancer associated microRNA. For more clarity the first sentence is modified and marked in yellow.

  1. Line 100: PNA and Aptamers are not biomarkers, they are used as biorecognition elements.

Response: The line has been corrected in the manuscript.

  1. Line 171: the title of the section should be renamed to “Detection of protein biomarkers”. Protein—based means the biorecognition element is a protein.

Response: The title of the section has been renamed as suggested.

  1. Line 185 and all over: CEA detection is based on antibodies but is in “protein” section. The overall classification of the paper is not very clear. The paper should be classified by in only one of the ways I think:

By biorecognition element: antibody, enzyme, nucleic acid probes (for hybridization), aptamers (both nucleic acid-based and peptide-based)

By the biomarker nature being detected: ions, proteins, nucleic acid (genes, RNAs), antibodies (autoantibodies), cells (microbial, cancer etc.)

By electrochemical biosensor types (DPV, EIS etc)

Right now the classification is messy. 3.1. discusses biosensors where DNA was used as a biorecognition element.

3.2 (Protein) discusses detection of

- protein biomarkers

- exosomes which are not proteins but are more complex structures

- cells!

For most of these analytes biorecognition elements are antibodies and sometimes aptamers. So by “protein” do the authors mean analyte (then why discuss cells and exosomes?) or biorecognition element (then why discuss aptamers?)

3.3. discusses detection of proteins by biosensors with antibodies as biorecognition elements.

So what is the difference between sections 3.2. where antibodies detected proteins and 3.3. is not clear to me.

Response: Necessary changes have been incorporated in the manuscript. 

  1. Line 356: the word “different” can be termed as “multiple” or “more than one” perhaps.

Response: The word “different” is renamed as “multiple” and marked in yellow.

Abbreviation of terms and typos:

  1. This has to be done in the very first mentioning of the term

Response: Necessary changes have been incorporated in the manuscript.

  1. This should not be repeated multiple times throughout the paper as it is done for instance for PSA.

Response: Necessary changes have been incorporated in the manuscript.

  1. Some terms used only once should not be abbreviated

Response: Necessary changes have been incorporated in the manuscript.

  1. CB – better to leave this as cancer biomarker without abbreviation (this is my personal opinion)

Response: Corrected as suggested

  1. Consistency of use should be kept. In one place Apt stands for aptamer while in others it is used simply as aptamers.

Response: Corrected as suggested

  1. Please correct H2O and H2O2

Response: Corrected

Need more extension in the following:

  1. Lines 52-55: What are the actual limitations of the assays?

Response: The limitations have been added and marked in yellow.

Reviewer 2 Report

The authors of this review "Current updates on cancer Biomarkers: Emphasizing therapy benefits" address the problem of tumor biomarkers not from the point of view of identifying new biomarkers but from the equally interesting point of view of the techniques to detect them. In fact, having very sensitive and low-cost methods available to detect these biomarkers early allows us to have more chances of healing or longer survival time. The authors describe in an exhaustive and clear way (the figures shown help to better understand the different detections) all the electrochemical biosensors techniques involved in detection of different cancer biomarkers used up to now in the research world. A single objection: the title does not seem congruent with the theme of the review because it does not mention the techniques of biosensors.

The authors of this review "Current updates on cancer Biomarkers: Emphasizing therapy benefits" address the problem of tumor biomarkers not from the point of view of identifying new biomarkers but from the equally interesting point of view of the techniques to detect them. In fact, having very sensitive and low-cost methods available to detect these biomarkers early allows us to have more chances of healing or longer survival time. The authors describe in an exhaustive and clear way (the figures shown help to better understand the different detections) all the electrochemical biosensors techniques involved in detection of different cancer biomarkers used up to now in the research world. A single objection: the title does not seem congruent with the theme of the review because it does not mention the techniques of biosensors.

Author Response

Comment:

The authors of this review "Current updates on cancer Biomarkers: Emphasizing therapy benefits" address the problem of tumor biomarkers not from the point of view of identifying new biomarkers but from the equally interesting point of view of the techniques to detect them. In fact, having very sensitive and low-cost methods available to detect these biomarkers early allows us to have more chances of healing or longer survival time. The authors describe in an exhaustive and clear way (the figures shown help to better understand the different detections) all the electrochemical biosensors techniques involved in detection of different cancer biomarkers used up to now in the research world. A single objection: the title does not seem congruent with the theme of the review because it does not mention the techniques of biosensors.

Response:

As suggested the title has been modified “Current update on Biomarkers for Early detection of Cancer: Comprehensive analysis’.

Reviewer 3 Report

Comment #1.

Although the title of this article is “Emphasizing Therapy Benefits”, it seems there is no mention or emphasis on therapeutic content within the paper. Please provide additional information regarding “therapy benefits” in the article or add a separate section to explain.

Comment #2.

In the abstract of the paper, the authors stated that they provide a comprehensive analysis of electrochemical biosensors involved in the detection of various cancer biomarkers. However, it seems that most of the main text lists the contents of the Table 1. rather than the author’s analysis. Please provide additional details about the author’s analysis.

Comment #3.

(page 7, line 253) Since the study mentioned here states that mi-RNA 155 and mi-RNA 122 are detected as biomarkers, I think it is more appropriate to be in ‘3.1. DNA based biomarker detection’ than in ‘3.2. Protein-based detection of biomarkers’.

Comment #4.

(page 13, line 401) In order to match the content and meaning of the abstract, I think the following corrections are necessary.

We feel that developing sensor would not facilitate early diagnosis but also enable us to manage cancer more effectively.

-> We feel that developing sensor would not only facilitate early diagnosis but also enable us to manage cancer more effectively.

Author Response

Comment #1.

Although the title of this article is “Emphasizing Therapy Benefits”, it seems there is no mention or emphasis on therapeutic content within the paper. Please provide additional information regarding “therapy benefits” in the article or add a separate section to explain.

Response: The title has been modified as per reviewer’s suggestion.

‘Current update on Biomarkers for Early detection of Cancer: Comprehensive analysis’.

 Comment #2.

In the abstract of the paper, the authors stated that they provide a comprehensive analysis of electrochemical biosensors involved in the detection of various cancer biomarkers. However, it seems that most of the main text lists the contents of the Table 1. rather than the author’s analysis. Please provide additional details about the author’s analysis.

Response: The manuscript describes about the different developed biosensing platforms for cancer biomarkers detection and the same has been described in Table 1 also.

Comment #3.

(page 7, line 253) Since the study mentioned here states that mi-RNA 155 and mi-RNA 122 are detected as biomarkers, I think it is more appropriate to be in ‘3.1. DNA based biomarker detection’ than in ‘3.2. Protein-based detection of biomarkers’.

Response: As suggested necessary changes have been made and marked in yellow in the original manuscript.

 Comment #4.

(page 13, line 401) In order to match the content and meaning of the abstract, I think the following corrections are necessary.

“We feel that developing sensor would not facilitate early diagnosis but also enable us to manage cancer more effectively.”

-> “We feel that developing sensor would not only facilitate early diagnosis but also enable us to manage cancer more effectively.”

Response: The changes have been marked in yellow.

Round 2

Reviewer 1 Report

The current title suits the paper content better than the previous one. However, the word “early” in the title suggest more content related to earliness of cancer detection to be added in the text. Authors might consider emphasizing this side of the paper more according to the title. What biomarkers among the discussed ones are important for early cancer diagnosis? There are number of references the authors site that are early biomarkers and these should be discussed more in this paper to reflect the title of the paper. Otherwise no need to write “early” as a buzzword.

Lines 24-28. Cancer definition should be (if made at all) improved I think. The current definition seems too simplified. Maybe better to show some up-to-date statistics on cancer or state that that it is still an issue despite progress in diagnosis and treatment etc.

Lines 57-58. Still, the limitations of the assays should be elaborated I think.

Lines 75-76. I don’t think the given classification is according to biological response

Lines 79-80. Secretary protein biomarkers should be replaced with corresponding antigens

Figure 1. Still does not make sense to me. Better to put as layers:

- biosensor types (optical etc)

-  ligands (aptamer etc

- HER2

Table 1. Please provide explanation for abbreviations such as PEDOT, PDITC below the table. No need to put both  the abbreviation and explanation in the table I think such as in: indium tin oxide (ITO). Provide only one.

Abbreviations: still some abbreviations are explained more than once such as LOD, AuNP, DPV

Section 3.2. is out of classification of other sections. Other sections are based on ligand used in the biosensors. Section 3.2. is about detecting protein biomarkers. But if you look closely, detection of protein biomarkers is seen throughout other sections (Section 3.3. and 3.3.) Better to put its content into other sections according to the ligands used in the studies. 

Author Response

Comment 1: The current title suits the paper content better than the previous one. However, the word “early” in the title suggests more content related to earliness of cancer detection to be added in the text. Authors might consider emphasizing this side of the paper more according to the title. What biomarkers among the discussed ones are important for early cancer diagnosis? There are number of references the authors site that are early biomarkers and these should be discussed more in this paper to reflect the title of the paper. Otherwise no need to write “early” as a buzzword.

Response: The title of the manuscript has been modified to “Current update on Biomarkers for detection of Cancer: Comprehensive analysis”.

Comment 2: Lines 24-28. Cancer definition should be (if made at all) improved I think. The current definition seems too simplified. Maybe better to show some up-to-date statistics on cancer or state that that it is still an issue despite progress in diagnosis and treatment etc.

Response: Necessary changes have been incorporated in the manuscript.

Comment 3: Lines 57-58. Still, the limitations of the assays should be elaborated I think.

Response: Necessary changes have been incorporated in the manuscript.

Comment 4: Lines 75-76. I don’t think the given classification is according to biological response

Response: Necessary changes have been incorporated in the manuscript.

Comment 5: Lines 79-80. Secretary protein biomarkers should be replaced with corresponding antigens

Response: Necessary changes have been incorporated in the manuscript.

Comment 6: Figure 1. Still does not make sense to me. Better to put as layers:

- biosensor types (optical etc)

-  ligands (aptamer etc

- HER2

Response: Figure 1 has been removed.

Comment 7: Table 1. Please provide explanation for abbreviations such as PEDOT, PDITC below the table. No need to put both the abbreviation and explanation in the table I think such as in: indium tin oxide (ITO). Provide only one.

Response: Necessary changes have been incorporated in the manuscript.

Comment 8: Abbreviations: still some abbreviations are explained more than once such as LOD, AuNP, DPV

Response: Necessary changes have been incorporated in the manuscript.

Comment 9: Section 3.2. is out of classification of other sections. Other sections are based on ligand used in the biosensors. Section 3.2. is about detecting protein biomarkers. But if you look closely, detection of protein biomarkers is seen throughout other sections (Section 3.3. and 3.3.) Better to put its content into other sections according to the ligands used in the studies

Response: Necessary changes have been incorporated in the manuscript.

Reviewer 3 Report

Comment #1. The author changed the title to “Current update on Biomarkers for Early detection of Cancer : Comprehensive analysis”. However, although the latest methods for detecting cancer biomarkers are introduced in this text, there is insufficient evidence and explanation regarding “early diagnosis(or detection)”. Therefore, according to the new title, clinical evidence of early diagnosis(or detection) and its comprehensive analysis should be presented in the text as well.

Comment #2. (page 8, line 311) misprint:  94] -> [94]

Author Response

Comment #1. The author changed the title to “Current update on Biomarkers for Early detection of Cancer: Comprehensive analysis”. However, although the latest methods for detecting cancer biomarkers are introduced in this text, there is insufficient evidence and explanation regarding “early diagnosis (or detection)”. Therefore, according to the new title, clinical evidence of early diagnosis (or detection) and its comprehensive analysis should be presented in the text as well.

 Response: The title of the manuscript has been modified to “Current update on Biomarkers for detection of Cancer: Comprehensive analysis”.

Comment #2. (page 8, line 311) misprint:  94] -> [94]

Response: Necessary changes have been incorporated in the manuscript.
